# Dose-Dependent Effect of Melatonin on BAT Thermogenesis in Zücker Diabetic Fatty Rat: Future Clinical Implications for Obesity

**DOI:** 10.3390/antiox11091646

**Published:** 2022-08-25

**Authors:** Samira Aouichat, Enrique Raya, Antonio Molina-Carballo, Antonio Munoz-Hoyos, Abdelkarim Saleh Aloweidi, Ehab Kotb Elmahallawy, Ahmad Agil

**Affiliations:** 1Department of Pharmacology, Biohealth Institute Granada (IBs Granada) and Neuroscience Institute, School of Medicine, University of Granada, 18016 Granada, Spain; 2Department of Rheumatology, University Hospital Clinic San Cecilio, 18016 Granada, Spain; 3Department of Pediatrics, School of Medicine, University of Granada, 18016 Granada, Spain; 4Unit of Pediatric Neurology and Neurodevelopment, University Hospital Clinic San Cecilio, Andalusian Health Service, 18016 Granada, Spain; 5Department of Anesthesia and Intensive Care, Faculty of Medicine, University of Jordan, Amman 11942, Jordan; 6Department of Zoonoses, Faculty of Veterinary Medicine, Sohag University, Sohag 82524, Egypt

**Keywords:** melatonin, dose-response, BAT, thermogenesis, UCP1, mitochondria, obesity, ZDF rat

## Abstract

Experimental data have revealed that melatonin at high doses reduced obesity and improved metabolic outcomes in experimental models of obesity, mainly by enhancing brown adipose tissue (BAT) thermogenesis. A potential dose-response relationship has yet to be performed to translate these promising findings into potential clinical therapy. This study aimed to assess the effects of different doses of melatonin on interscapular BAT (iBAT) thermogenic capacity in Zücker diabetic fatty (ZDF) rats. At 6 wk of age, male ZDF rats were divided into four groups (*n* = 4 per group): control and those treated with different doses of melatonin (0.1, 1, and 10 mg/kg of body weight) in their drinking water for 6 wk. Body weight (BW) was significantly decreased at doses of 1 and 10 mg/kg of melatonin, but not at 0.1 mg/kg compared with the control, with a similar rate of BW decrease being reached at the dose of 1 mg/kg (by ~11%) and 10 mg/kg (by ~12%). This effect was associated with a dose-dependent increase in the thermal response to the baseline condition or acute cold challenge in the interscapular area measurable by infrared thermography, with the highest thermal response being recorded at the 10 mg/kg dose. Upon histology, melatonin treatment markedly restored the typical brownish appearance of the tissue and promoted a shift in size distribution toward smaller adipocytes in a dose-dependent fashion, with the most pronounced brownish phenotype being observed at 10 mg/kg of melatonin. As a hallmark of thermogenesis, the protein level of uncoupled protein 1 (UCP1) from immunofluorescence and Western blot analysis increased significantly and dose-dependently at all three doses of melatonin, reaching the highest level at the dose of 10 mg/kg. Likewise, all three doses of melatonin modulated iBAT mitochondrial dynamics by increasing protein expression of the optic atrophy protein type 1 (OPA1) fusion marker and decreasing that of the dynamin-related protein1 (DRP1) fission marker, again dose-dependently, with the highest and lowest expression levels, respectively, being reached at the 10 mg/kg dose. These findings highlight for the first time the relevance of the dose-dependency of melatonin toward BW control and BAT thermogenic activation, which may have potential therapeutic implications for the treatment of obesity. To clinically apply the potential therapeutic of melatonin for obesity, we consider that the effective animal doses that should be extrapolated to obese individuals may be within the dose range of 1 to 10 mg/kg.

## 1. Introduction

The prevalence of obesity has dramatically increased over the past 40 years and is expected to increase further to ~30.3 billion by 2030 [1]. Obesity develops when energy intake chronically exceeds energy expenditure (EE), with the deposition of this excess energy primarily as triglycerides in white adipose tissue [2]. Most of the current available anti-obesity pharmacological approaches act by reducing caloric intake or impairing fat absorption [3]. However, because these approaches are associated with multiple side effects [3], safe and effective alternatives for treating and overcoming obesity are urgently needed. In recent years, targeting BAT thermogenesis to increase EE has emerged as a promising therapeutic strategy to combat obesity [4].

Unlike white adipose tissue, which is involved in storing excess energy as fat that can be mobilized in times of need, BAT can convert stored fat into heat energy through a process termed adaptive non-shivering thermogenesis, which plays an important role in temperature regulation in rodents and newborn babies [5]. Morphologically, BAT is composed of multilocular lipid droplets and high contents of mitochondria that contain a specific protein in the inner mitochondrial membrane called uncoupling protein 1 (UCP1) [6]. The UCP1 plays a critical role in BAT-mediated thermogenesis by causing a proton gradient leak across the mitochondrial membrane and thereby uncoupling oxidative phosphorylation to generate heat instead of ATP synthesis [7]. The UCP1 in brown adipocytes is induced by a wide range of metabolite sources, among them fatty acids and glucose, which are major contributors and play a pivotal role in BAT thermogenesis [5,8]. Activated BAT can use glucose and fatty acids to maintain a high oxidative metabolic capacity, and when it is highly active, it exerts beneficial metabolic effects [5,8]. The oxidative metabolism and uncoupled respiration in brown adipocytes are mainly accomplished by mitochondria, which hold a central position in thermogenesis [5]. These organelles undergo dynamic processes of fusion and fission that can serve as a mechanism for metabolic adaptation to mitochondrial bioenergetic efficiency and energy substrate oxidation [5]. Therefore, normal mitochondrial function is essential for maintaining BAT function and metabolic homeostasis.

Recently, BAT research has dramatically increased after the rediscovery of metabolically active BAT in adult humans using ^18^fluoro-2-deoxyglucose positron emission tomography (^18^FDG-PET). Importantly, the prevalence and activity of adult human BAT, as assessed by 18FDG-PET, have been found to correlate negatively with body mass index (BMI) in humans [9,10,11,12,13,14], and individuals with detectable metabolically active BAT have a lower risk of developing type 2 diabetes and heart diseases [15]. It is therefore expected that targeting BAT thermogenesis could be a promising strategy and a hopeful target for counteracting obesity and related metabolic disorders. In support of this perspective, an earlier prospective cohort study has shown an increased prevalence of BAT activity after weight loss in morbidly obese patients undergoing gastric banding surgery, suggesting the effectiveness of targeting BAT activation in humans as an anti-obesity therapy [16]. Several candidates are now being developed as possible therapeutic targets in the process of BAT activation, but their potential adverse or toxic effects may limit their long-term use for this purpose [3,17]. In this context, there is an urgent and highly demanded need for developing effective and safe new drugs to reduce obesity by targeting BAT. In recent decades, melatonin has attracted increasing attention as a promising therapeutic for obesity and related diseases mainly due to its high efficacy and remarkable lack of toxicity and side effects at any dose, even for long-term use in humans [18,19].

Melatonin (N-acetyl-5-methoxytryptamine) is a naturally occurring substance mainly produced and secreted from the pineal gland and is well known for its role in controlling the circadian rhythm [20]. Exogenous melatonin is widely used to remedy sleep disorders and jet lag, either as a dietary supplement or as a drug, in many European countries and the USA [21,22]. The extraordinary intrinsic properties of melatonin, including high cell permeability, the ability to reach subcellular compartments, and its role as a robust anti-inflammatory, antioxidant and free radical scavenging, and anti-apoptotic agent, as well as its acceptable safety profile, make it a promising approach for the treatment of various diseases, including obesity and related diseases [23]. There is growing evidence that exogenous melatonin exerts beneficial effects on obesity and related metabolic disorders in animals; nevertheless, clinical evidence of its effectiveness in obese individuals is still insufficient and conflicting [4], which makes it difficult to draw a firm conclusion and thus further clinical trials are warranted to clarify this issue. The BAT thermogenic mechanism has been proposed to account for the anti-obesogenic effect of melatonin [24,25,26]. In this context, our previous studies carried out in ZDF rats have shown that melatonin treatment exhibited promising effects in increasing BAT mass and thermogenic activity, increasing UCP1 expression, improving mitochondrial respiration, and reducing brown adipocyte susceptibility to mitochondrial oxidative/nitrosative stress and apoptosis and, in addition, prevented BW gain and glucose–lipid metabolic disorders [26,27,28,29]. Although the observed beneficial effects of melatonin on BAT have not been translated yet to clinical experimentation, a small-scale proof of concept study in patients with melatonin deficiency due to radiotherapy or surgical removal of the pineal gland has shown that melatonin replacement therapy increased BAT volume and activity and improved cholesterol and triglyceride blood levels [30].

On the above basis, and given the ability of BAT to dissipate energy as heat and to contribute to EE, we are optimistic that melatonin can be used as a BAT activator for future therapeutic applications in obesity. Thus, information on the potentially effective dose of melatonin is urgently needed to underscore the clinical meaning of this potential therapy for obesity treatment in the future. To this end, the present study aimed to perform an effective dose-setting study to evaluate the effects of three different doses of melatonin (0.1, 1, and 10 mg/kg) on iBAT thermogenic capacity in ZDF rats. The determination of the dose-response relationship will have important implications on dosing regimens in clinical application, not only to determine the effective dose that produces the desired therapeutic effect but also because of its implications for economical and safety aspects, since lower doses may have the same desired therapeutic endpoint.

## 2. Materials and Methods

### 2.1. Ethical Statement

Ethical approval of this study was obtained from the Ethical Committee at the University of Granada (Granada, Spain) under the reference number 4-09-2016-CEEA, according to the European Union guidelines for animal care and protection.

### 2.2. Reagents

All reagents used were of the highest purity available. Melatonin was purchased from Sigma-Aldrich (Madrid, Spain).

### 2.3. Animals and Experimental Protocol

Male Zücker diabetic fatty rats (ZDF; fa/fa) were obtained at 5 wk of age from Charles River Laboratory (Charles River Laboratories, SA, Barcelona, Spain). Animals were maintained on Purina 5008 rat chow (protein 23%, fat 6.5%, carbohydrates 58.5%, fibre 4%, and ash 8%; Charles River Laboratories, SA, Barcelona, Spain) and housed two per clear plastic cage under a 12 h light/dark cycle (lights on at 07:00 a.m.) and controlled environmental conditions of temperature (28–30 °C) and relative humidity (30–40%). The animals were acclimated to room conditions for one week before the experiments, and water intake was recorded.

Upon initiation of the experiment, the animals were randomly divided into four groups (n = 4 per group) as follows: the first three groups were treated with melatonin in their drinking water at the doses of 0.1, 1, and 10 mg/kg b.w/day (Mel-0.1, Mel-1, and Mel-10, respectively) for 6 consecutive weeks, and the remaining group served as a vehicle-treated control (Control). Melatonin was dissolved in a minimum volume of absolute ethanol and then diluted in the drinking water to yield doses of 0.1, 1, and 10 mg/kg b.w, with a final ethanol concentration of 0.066% (*w*/*v*). The control group received the vehicle only in the drinking water at a comparable concentration and treatment duration. Fresh melatonin and vehicle solutions were prepared three times a week, and melatonin doses were adjusted for BWs throughout the experimental period. To carefully determine the treatment dose throughout the experimental period, water intake and BWs were recorded three times a week. All water bottles were covered with aluminium foil to prevent melatonin photodegradation. The BW of all animals was recorded at the beginning and the end of the experiment. At the end of the experiment, the animals were sacrificed under sodium thiobarbital (thiopental) anaesthesia, and the interscapular brown fat pads of each rat were immediately dissected and then frozen at −80 °C for further analysis. A portion of the iBAT pads was fixed in 4% paraformaldehyde for histological analysis.

### 2.4. Infrared Thermal Imaging Measurments

After overnight fasting, ZDF rats from the four groups were either kept at 28–30 °C (room temperature) or exposed to 4 °C for 5 min for an acute cold challenge between 09:00 and 11:00 h. The acute cold challenge was performed by placing the rat on a hot/cold plate analgesia meter precooled to 4 °C (Panlab SLU, Barcelona, Spain).

Thermal images of the BAT in the interscapular region were recorded with a thermal imaging camera (FLIR B425, FLIR Systems AB, Danderyd, Sweden) with a range limit of −20 °C to 120 °C. Dorsal thermal images at a perpendicular distance of 20 cm were regularly captured before (at room temperature) and immediately after the acute cold challenge. The images were analysed with a specific software package (FLIR Systems software).

### 2.5. Purification of Mitochondria

Mitochondrial protein extract was isolated from iBAT by serial centrifugation according to the instructions previously described by our research group [31]. Briefly, about 300 mg of iBAT was washed with cold saline and homogenized (1:10 *w/v*) in an isolation medium (10 mM Tris, 250 mM sucrose, 0.5 mM Na_2_EDTA, and 1 g/L free fatty acid bovine serum albumin (BSA); pH 7.4; 4 °C) with a Teflon pestle. The homogenate was centrifuged at a low speed (1000× *g*) for 10 min at 4 °C, and the resultant supernatant was decanted to fresh tubes and subjected to high-speed centrifugation (15,000× *g*) for 20 min at 4 °C. The resultant pellet containing the mitochondria was resuspended in 1 mL of isolation medium (BSA free) and centrifuged again at high speed (15,000× *g*) for 20 min at 4 °C to pellet the pure mitochondria. The pelleted pure mitochondria were resuspended in 1 mL of respiration buffer (20 mM HEPES, 0.5 mM EGTA, 3 mM MgCl_2_, 20 mM taurine, 10 mM KH_2_PO_4_, 200 mM sucrose, and 1 g/L fatty-acid-free BSA) and incubated on ice for 10–15 min to allow the rearrangement of the membranes. Aliquots of pure mitochondrial extracts were stored at −80 °C for Western blotting. Protein concentration in the mitochondrial extracts was measured by the Bradford method [32] using BSA as a standard.

### 2.6. Western Blot Analysis

Western blot analysis was performed according to the instructions previously described by our research group [31]. Equal amounts of mitochondrial protein extracts were resolved on SDS-PAGE (sodium dodecyl sulphate polyacrylamide gel electrophoresis). The gels for immunoblot analyses were transferred to a nitrocellulose membrane (Bio-Rad Trans-Blot SD, Bio-Rad Laboratories). The membranes were then blocked with 5% non-fat dry milk in tris-buffered saline (TBS) containing 0.05% Tween-20 (TBS-T) for 1 h at 37 °C and incubated overnight at 4 °C with primary antibodies against UCP1 (cat#U-6382), OPA1 (cat#SAB-5700860), and DRP1 (cat#SAB-5700783) at 1:500–1:2000 dilutions. All the antibodies were obtained from Sigma-Aldrich (Sigma-Aldrich, Madrid, Spain). Equal protein loading was demonstrated by incubating the membranes with mouse β-actin antibody (cat#SC-81178; Santa Cruz Biotechnology, Santa Cruz, CA, USA) at 1:1000 dilution. After the incubation, the membranes were washed three times for 20 min in TBS-T and incubated for 1 h at room temperature with respective horseradish-peroxidase-conjugated secondary antibodies (Sigma-Aldrich, Madrid, Spain) at 1:1000 dilution. The membrane was washed three times for 20 min in the TBS-T, and then a chemiluminescence assay system (ECL kit, GE Healthcare Life Sciences, Buckinghamshire, UK) was used to develop the immunoreactivity bands. Finally, the protein band densities were quantitatively analysed using Image J 1.33 software (National Institutes of Health, Bethesda, MD, USA). The results were normalized to β-actin as a loading control. All experiments were performed in triplicate.

### 2.7. Histological Analysis

Following anaesthesia, excised interscapular brown fat was fixed overnight in 10% (*w*/*v*) buffered formaldehyde at 4 °C for 24 h, rinsed with 0.1 M phosphate buffer (PBS (phosphate-buffered saline) pH 7.4, dehydrated in a graded series of ethanol (from 80% to absolute alcohol), cleared in xylene, and embedded in paraffin. The paraffin blocks from each group were cut with a microtome into 4 μm-thick sections, stained with haematoxylin and eosin (H&E), and inspected under a light microscope (Olympus, Germany) equipped with a digital camera system (Carl Zeiss camera, model Axiocam ERc 5s. Göttingen, Germany). Images of H&E-stained tissue sections were digitized, and adipocyte size and frequency distribution of adipocyte size was determined using Image J 1.33 software (National Institutes of Health, Bethesda, MD, USA). The average adipocyte size was expressed as the average cross-sectional area per cell (μm^2^/cell) of the tissue sample, which was calculated based on the values of about 200 random adipocytes per group.

### 2.8. Immunofluorescence Staining for UCP1

For UCP1 immunofluorescence, unstained deparaffinised sections were treated with citrate buffer pH 6.0 for 20 min at 60 °C for antigen retrieval and then blocked with 5% BSA, followed by overnight incubation at 4 °C with the same UCP1 antibody used for Western blot analysis (cat# U-6382; Sigma-Aldrich, Madrid, Spain) at 1:1000 dilution. After incubation, the tissue sections were washed three times with a 1 × PBS solution containing Tween-20 and then incubated with secondary anti-rabbit cyanine3 antibody (Cy3) (cat# A-10520; Invitrogen, Molecular Probes, Carlsbad, CA, USA). Fluorescence photomicrographs were captured by an Olympus IX2 fluorescence microscope at a magnification of ×200. All photomicrographic images from each group were captured under the same camera and microscope settings. The immunofluorescence intensities were quantitatively analysed using Image J 1.33 software (National Institutes of Health, Bethesda, MD, USA).

### 2.9. Statistical Analysis

Statistical Package of Social Science (IBM SPSS Software, version 15, Michigan, IL, USA) was used for statistical analysis. All results are expressed as mean ± standard deviation (S.D.) values. Comparisons between experimental groups were analysed using one-way ANOVA followed by the Tukey post hoc test. Differences between group means were considered statistically significant if *p* < 0.05.

## 3. Results

### 3.1. Dose-Dependent Effects of Melatonin on Body Weight and iBAT Thermogenic Activity

We have previously shown that melatonin at a dose of 10 mg/kg for 6 wk protected against the excessive BW gain and increased the thermogenic function of iBAT in the ZDF rats [26,27]. Here, we investigated whether 0.1, 1, and 10 mg/kg of melatonin for the same treatment period and in the same animal strain showed dose-dependent effects on BW and iBAT thermogenic activity.

After 6 wk of treatment, melatonin significantly decreased the BW in the Mel-1 (by ~11%) and Mel-10 (by ~12%) groups compared with the control group (*p* < 0.05 and *p* < 0.05, respectively; Table 1), with no significant difference among them (*p* > 0.05; Table 1). In the Mel-0.1 group, the final BW showed a trend towards a nonsignificant decrease (by ~5%) compared with the control group (*p* > 0.05; Table 1). The final BW was found to be significantly lower in the Mel-10 and Mel-1 groups than in the Mel-0.1 group (*p* < 0.05 and *p* < 0.05, respectively; Table 1).

To assess the degree of iBAT thermogenic activity, we first measured the interscapular skin temperature by infrared thermography at both baseline and acute cold conditions. The acute cold test was performed to gauge the adaptive thermogenesis in iBAT, which has been found to be blunted in obese and overweight individuals in the cold-activated state [33,34]. The interscapular fat depot was selected because it is the largest and most accessible BAT depot in rodents that may correspond to the supraclavicular BAT depot described in adult humans [35].

Based on the experiment, the baseline interscapular skin temperature was significantly higher in the Mel-1 (35.4 ± 0.1 °C) and Mel-10 (35.8 ± 0.2 °C) groups than in the control group (34.3 ± 0.2 °C; *p* < 0.05 and *p* < 0.05, respectively; Figure 1a), without a significant difference among them *(p* > 0.05; Figure 1a). There was no significant difference in the baseline interscapular skin temperature between the Mel-0.1 (34.5 ± 0.1 °C) and control groups (*p* > 0.05; Figure 1a). Significantly higher interscapular skin temperature degrees were found in the Mel-10 and Mel-1 groups than in the Mel-0.1 group (*p* < 0.05 and *p* < 0.05, respectively; Figure 1a).

After the cold challenge, the interscapular skin temperature rose significantly in the Mel-1 (0.26 ± 0.04 °C) and Mel-10 groups (0.38 ± 0.05 °C) in comparison with the control group (0.04 ± 0.03 °C; *p* < 0.05 and *p* < 0.01, respectively; Figure 1b), with no significant difference among them *(p* > 0.05; Figure 1b). A non-significant trend towards a higher increase in the interscapular skin temperature was observed in the Mel-0.1 group (0.10 ± 0.01 °C) when compared with the control group *(p* > 0.05; Figure 1b). The Mel-10 and Mel-1 groups displayed a significantly higher increase in the interscapular skin temperature than the Mel-0.1 group (*p* < 0.05 and *p* < 0.05, respectively; Figure 1b). The dose-dependent effect of melatonin treatment on the iBAT thermogenesis was also clearly apparent in the infrared thermal images shown in Figure 1c.

### 3.2. Dose-Dependent Effects of Melatonin on iBAT Morphology

To explore whether the dose-dependent effect of melatonin on iBAT thermogenesis was reflected in particular morphological characteristics, iBAT histological analysis was performed. As shown in the H&E-stained preparations (Figure 2a), the iBAT of the ZDF control rats consisted predominately of hypertrophic unilocular adipocytes with a large lipid droplet, giving a white-like appearance rather than the typical brownish phenotype. Consistent with the above-described infrared thermography results, all three doses of melatonin resulted in a gradual suppression of unilocular lipid droplet deposition, with the smallest sign of whitening being observed in the Mel-10 group. Notably, an apparent accumulation of smaller adipocytes with multilocular lipid droplets was observed in the iBAT of the Mel-1 and Mel-10 groups. In the Mel-0.1 group, although the brownish phenotype of the iBAT was not evident, it was clearly distinguishable from the control iBAT, with apparently smaller unilocular adipocytes.

The morphometric quantitative analysis shows that adipocyte size was significantly smaller in the Mel-1 (420.4 ± 83.9 μm^2^) and Mel-10 (425.1 ± 95.0 μm^2^) groups than the control group (926.7 ± 109.1 μm^2^; *p* < 0.05 and *p* < 0.05, respectively; Figure 2b), with no significant difference among them (*p* > 0.05; Figure 2b). A non-significant tendency toward smaller adipocyte size was appreciated in the Mel-0.1 group (711.2 ± 93.8 μm^2^) compared with the control group (*p* > 0.5; Figure 2b). The Mel-1 and Mel-10 groups exhibited significantly smaller adipocytes than the Mel-0.1 group (*p* < 0.05 and *p* < 0.05, respectively; Figure 2b). In addition, the frequency distribution analysis shows a shift in size distribution toward smaller adipocytes in all three melatonin-treated groups (Figure 2c). Notably, a significant and progressive decreased proportion of larger (800–1100 μm^2^) and increased proportion of smaller (200–500 μm^2^) adipocytes were noted in iBAT from the Mel-0.1 (larger: 40%; smaller: 15%), Mel-1 (larger: 20%; smaller: 45%), and Mel-10 (larger: 10%; smaller: 65%) groups, compared with the control group (*p* < 0.05, *p* < 0.01 and *p* < 0.001, respectively; Figure 2c).

### 3.3. Dose-Dependent Effects of Melatonin on Thermogenic and Mitochondrial Dynamic Markers

To characterize the dose-dependent effects of melatonin treatment on the iBAT thermogenic activity, we next assessed the thermogenic UCP1 protein expression in the iBAT using immunofluorescence and Western blotting. UCP1 is known as a key molecule for BAT thermogenesis [36].

By immunofluorescence assay, UCP1 protein expression showed weakened fluorescent immunoreactivity in the iBAT of the control group, whereas melatonin treatment progressively increased the UCP1 fluorescent immunoreactivity with increasing doses of melatonin and achieved the strongest fluorescent signal at the 10 mg/kg dose (Figure 3a). The quantitative analysis of the UCP1 fluorescent immunoreactivity intensity indicated significantly higher UCP1 expression in all melatonin-treated groups, Mel-0.1 (3.4-fold), Mel-1(8.0-fold), and Mel-10 (13.2-fold), relative to the control group (*p* < 0.05, *p* < 0.01 and *p* < 0.001, respectively; Figure 3b). The highest UCP1 immunoreactivity intensity was observed in the Mel-10 group, as compared with the Mel-1 (1.6-fold; *p* < 0.05; Figure 3b) and Mel-0.1 (3.9-fold; *p* < 0.01; Figure 3b) groups. The Mel-1 group showed a significantly higher UCP1 immunoreactivity than the Mel-0.1 group (2.3-fold; *p* < 0.05; Figure 3b).

Western blotting of UCP1 in the mitochondrial fractions of iBAT was also performed to further support the above findings. The protein level of UCP1 determined by Western blot was considered the most relevant parameter to estimate the BAT thermogenic capacity in response to chronic stimuli since it correlates quantitatively and temporally with the total thermogenic capacity [37]. Consistent with the results of the immunofluorescence, Western blot analysis indicated that the protein expression of UCP1 is also significantly upregulated in all melatonin-treated groups, Mel-0.1 (3.3-fold), Mel-1(6.4-fold), and Mel-10 (12.4-fold), as compared with that in the control group (*p* < 0.05, *p* < 0.05 and *p* < 0.001, respectively; Figure 3c). The highest protein level of UCP1 was observed in the Mel-10 group, as compared with the Mel-1 (1.9-fold; *p* < 0.05; Figure 3c) and Mel-0.1 (3.8-fold; *p* < 0.01; Figure 3c) groups. The Mel-1 group showed a significantly higher UCP1 protein amount than the Mel-0.1 group (1.9-fold; *p* < 0.05; Figure 3c).

Mitochondrial fusion and fission dynamics play a critical role in BAT thermogenesis, and the abnormal function of mitochondria dynamics leads to obesity and related diseases [38,39,40]. Accordingly, we explored the possible dose-dependent effects of melatonin treatment on the iBAT mitochondrial dynamic by assessing OPA1 and DRP1 levels in isolated iBAT mitochondria by Western blotting. OPA1 mediates mitochondrial fusion, whereas DRP1 is considered a key regulator of mitochondrial fission [41].

As shown in Figure 3d, OPA1 protein level was found to be significantly upregulated in all melatonin-treated groups, Mel-0.1 (2.2-fold), Mel-1 (2.2-fold), and Mel-10 (5.5-fold), compared with that in the control group (*p* < 0.05, *p* < 0.05 and *p* < 0.01, respectively). The highest protein level of OPA1 was observed in the Mel-10 group, as compared with the Mel-1 (2.5-fold; *p* < 0.05) and Mel-0.1 (2.5-fold; *p* < 0.05) groups. There was no statistically significant difference between the Mel-0.1 and Mel-1 groups (*p* > 0.05) in the protein expression of OPA1.

In addition, the protein level of DRP1 was found to be significantly repressed in all melatonin-treated groups, Mel-0.1 (1.7-fold), Mel-1 (5.8-fold), and Mel-10 (3.6-fold), compared with that in the control group (*p* < 0.05, *p* < 0.01 and *p* < 0.01, respectively; Figure 3e), with no significant difference between the Mel-1 and Mel-10 groups (*p* > 0.5; Figure 3e). The Mel-1 and Mel-10 groups displayed a significantly lower DRP1 protein content than the Mel-0.1 group (3.5-fold and 2.1-fold; *p* < 0.05 and *p* < 0.05, respectively; Figure 3e).

## 4. Discussion

The present findings revealed for the first time that chronic daily oral administration of melatonin at doses of 1 and 10 mg/kg b.w, but not at a 0.1 mg/kg b.w dose, prevents BW gain and enhances the basal and adaptive iBAT thermogenesis in obese diabetic ZDF rats. At the molecular level, melatonin treatment showed successful dose-response effects on mitochondrial thermogenic and dynamic expression markers, even at a 0.1 mg/kg dose, suggesting a previously unrecognized potency and efficacity of melatonin in targeting mitochondria, which might have a valuable therapeutic role in the treatment of obesity.

The BW gain-lowering effect of melatonin has generated much interest in recent years and has been extensively studied in animal models and clinical trials. The data from animal studies using various models of obesity have convincingly shown that orally administered melatonin in doses ranging between 1 and 100 mg/kg b.w per day resulted in reduced BW gain [4,42,43,44]. Our present results confirm the existing evidence for the beneficial effect of melatonin on BW, showing that oral melatonin at doses of 1 and 10 mg/kg for 6 wk could subsequently reduce BW gain. Importantly, as our current data shows that 1 and 10 mg/kg of melatonin produces practically similar effects on the BW, it is reasonable to suggest that the dose of 1 mg/kg may be the maximum effective dose, which needs to be verified with an additional more precise effective dose setting study. If confirmed, an extrapolation of 1 mg/kg of melatonin would be sufficient to achieve maximum BW loss in obese individuals, which would be, on the one hand, of crucial importance to prevent unnecessary high dosing and, on the other hand, would be highly advantageous in terms of patient costs. In any case, since both doses were shown to be effective in reducing the BW gain, we estimate that the effective melatonin dose to achieve desirable effects on BW loss in ZDF rats is within the dose range of 1 to 10 mg/kg b.w. If we convert these animal doses to the human doses according to standard dose translation, based on dividing the surface area by factor 6.17, the calculated equivalent human dose is between 0.16 and 1.6 mg/kg per day (1/6.17 = 0.16 and 10/6.17 = 1.6) [45]. Thus, we anticipate that the therapeutically effective dose to achieve significant weight loss in obese individuals is within the dose range of 0.16 to 1.6 mg/kg of total BW per day. For an adult individual of 75 kg, further clinical trials using melatonin with doses in the 12 to 120 mg/day range (0.16 × 75 = 12 and 1.6 × 75 = 120) may be warranted and, in addition, appear to be reasonable since this dose range would not cause serious adverse events, based on the data of some clinical studies using melatonin in other health conditions [18,19,46,47,48,49,50,51,52]. Nevertheless, because there are fewer reports on the safety of high doses of melatonin (≥100 mg/day), its potential efficacity in obese patients should be closely screened for relevant safety concerns. This anticipated range of doses is far from that usually given in clinical trials for obesity. Indeed, the melatonin doses used in obese subjects have ranged from 1 to 10 mg per day orally, and the data from various clinical trials on the effects of melatonin on BW loss outcomes are conflicting, ranging from a modest to no effect, depending on the dose, duration of administration, and whether used as adjunct or single therapy [53,54,55,56,57]. It is worth mentioning that the positive findings from the clinical trials reported here should not be used as a confirmation of melatonin’s efficacity in weight loss since melatonin has been given as an adjunct to a low-calorie diet therapy that may confound the relationship between the melatonin and weight loss outcomes [53,54]. We are optimistic that the present dose-response data may help resolve the previous conflicting clinical findings of a possibly inadequate dose of melatonin. On the other hand, the important feature of the current study is that we used a relatively low dose of 0.1 mg/kg melatonin, far from the melatonin doses that have been usually offered to experimental animals. Compared with the control, such a melatonin dose failed to show any significant difference in BW over a 6 wk period. Given that this dose is equivalent to ~1.2 mg of melatonin for a 75 kg adult individual, this finding is consistent with a randomized clinical trial that has shown that daily oral administration of melatonin at a dose of 1 or 3 mg for one year in post-menopausal women weighing 66.6 –76.0 kg had no significant effect on BW and BMI compared with placebo [57].

Although the mechanism behind melatonin-induced BW reduction is unclear, increased BAT thermogenic activity to increase EE could account for the phenotype, as we proposed previously [26]. Since the rediscovery of functional BAT in adult humans, extensive studies have examined the role of active BAT in metabolism in human adult populations. Most clinical studies investigating BAT thermogenic capacity have been performed under cold challenge, which is well known as a powerful natural tool to detect BAT prevalence and activation [58]. In this scenario, using a standardized cooling protocol, several prospective studies have shown that obesity was associated with impaired cold-induced BAT activation, which could be indicative of a loss of BAT thermogenic capacity in obese subjects [9,16,33,34,59,60]. Coupled with this observation, defective BAT activities upon cold exposure have been reported in various experimental models of obesity [61]. Consistent with this observation, our previous study using infrared thermography to measure the temperature of the skin overlying the iBAT, has shown an impaired iBAT thermogenic response to acute cold challenge in ZDF rats and that melatonin at a dose of 10 mg/kg has been found to reverse this pattern to normal [26]. Our main goal in performing the cold challenge test was to gauge the thermogenic ability of iBAT and investigate whether it is restored after melatonin treatment since, at room temperature (baseline), the animals are chronically adapted to that temperature and have no physiological need to activate it. The present results confirm our previous data and, in addition, show an attractive dose-response relationship between melatonin treatment and cold-induced iBAT activation. Notably, melatonin at doses of 1 and 10 mg/kg resulted in a significant rise in the interscapular skin temperature, with the maximum effect being noticed at 10 mg/kg, whereas melatonin at a dose of 0.1 mg/kg did not seem to be as effective as the other doses. This finding would tend to indicate that melatonin restores obesity-induced iBAT thermogenic dysfunction, and this seems to be in a dose-dependent manner. In addition, even at baseline conditions, where the animals are adapted to that temperature and do not need higher heat production, melatonin at doses of 1 and 10 mg/kg, but not at 0.1 mg/kg, was found to increase the interscapular skin temperature, which may reflect the enhancement of the baseline BAT activity. The lack of a significant effect with a dose of 0.1 mg/kg of melatonin on interscapular skin temperature is related either to the short treatment duration (6 wk) or the relatively low melatonin dose, which may not be sufficient to produce a significant thermogenic response. Hence, further investigation with a prolonged duration of treatment is necessary before drawing any firm conclusion on this issue. These findings are in agreement with those of other studies showing that melatonin at doses ranging from 1 to 10 mg/kg plays a crucial role in the regulation of BAT thermogenesis in melatonin deficient experimental models and seasonal breeder species [62,63,64,65,66]. Notwithstanding that our iBAT temperature results are not original, they expand and advance previous results by supporting the potential use of melatonin to promote BAT thermogenesis in the obesity context in a dose-dependent manner.

The hallmark of dysfunctional BAT in obesity is a whitening phenotype that has been previously characterized in experimental models of obesity by enlarged lipid droplet accumulation, mitochondrial dysfunction, and functional loss [67,68,69,70]. Consistent with this notion, the iBAT from the control ZDF group assumed a striking whitening appearance, characterized by excessive accumulation of hypertrophied lipid droplets and pale eosinophilic staining, which strongly indicates fuel switching from thermogenesis to lipid storage. In line with this idea, the BAT whitening phenotype has also been described previously in the ZDF strain model, with impaired glucose uptake and an increase in the fatty acid synthesis enzyme [67]. We have previously shown that melatonin at a dose of 10 mg/kg successfully recovered the typical microscopic appearance of the iBAT, including depleted lipid content, multilocular adipocytes, and dense eosinophilic staining [26]. In the present study, 1 mg/kg of melatonin was found to be as effective as 10 mg/kg for recovering the typical iBAT appearance; however, the brownish microscopic appearance tends to be more pronounced at 10 mg/kg. In the case of 0.1 mg/kg of melatonin, the brownish phenotype of the tissue was not evident, but the adipocytes were remarkably smaller sized than the control. These results seem compatible with the infrared thermography results and thus support the idea that they might likely reflect the reversibility of iBAT thermogenic capacity in a dose-response manner. The reduced lipid deposition within the iBAT following the melatonin treatment most likely reflects increased fatty acid β-oxidation, given that mitochondrial fatty acid β-oxidation has been demonstrated to be critical for maintaining the brown adipocyte phenotype both during times of activation and quiescence (basal state) [71]. Fatty acid β-oxidation fuels the increase in uncoupled mitochondrial respiration and contributes to inducing the expression of thermogenic genes such as UCP1 [71]. UCP1 is a hallmark of brown adipocytes and plays a pivotal role in BAT thermogenic function since it confers to brown adipocytes their specific ability to dissipate the proton gradient as heat, and its downregulation or absence has been reported to be associated with impaired BAT thermogenic capacity and lower EE [72,73,74]. The efficacy of melatonin to induce UCP1 expression in iBAT has already been demonstrated by other groups in experimental ageing and melatonin deficient models, and the melatonin doses offered to animals have ranged from 1 to 10 mg/kg.bw [64,65,66,75]. This present study expanded and advanced the previous finding by showing that the iBAT UCP1 protein expression from immunofluorescence and Western blot analysis significantly increased at doses of 1 and 10 mg/kg of melatonin and, surprisingly, even at 0.1mg/kg of melatonin in an experimental model of diabetes and obesity. These findings argue again for a recovery of the iBAT thermogenic phenotype, which might be attributable to the recruitment and/or reactivation of BAT. The dose-response pattern of UCP1 protein expression in response to melatonin was consistent with that of the infrared thermal image data. Of note, UCP1 protein levels progressively increased with increasing doses of melatonin and reached the highest increase at a dose of 10 mg/kg, which could explain the highest iBAT temperature observed under acute cold exposure and basal state. Another point especially worthy of note, is that the significant increase in UCP1 level at a dose of 0.1 mg/kg of melatonin was not associated with changes in the iBAT temperature that were observed under both basal and cold conditions. This dissociation could be explained either by the fact that the amount of UCP1 encoded protein in the tissue at the dose of 0.1 mg/kg of melatonin was not sufficient enough to be translated to a significant iBAT thermogenic response or by the small number of animals used for the infrared thermograph test, as a larger number of the animals might be needed to demonstrate significant differences. In any case, this result might be advantageous since, on the one hand, it indicates that future clinical applications using this treatment for promoting the BAT thermogenic phenotype in obese individuals are less prone to any potential side effects and more cost-effective in economic terms. On the other hand, it would tend to indicate that the melatonin molecule is endowed with a higher potency than previously recognized.

One possible explanation for the BAT thermogenic repression in obese individuals could be related to the inflammatory and oxidative stress microenvironment in BAT that is known to occur in the adiposity state [61]. As previously reported in various obesity models, these deleterious conditions have been found to impair mitochondrial respiratory and antioxidant functions, which ultimately cause loss of BAT thermogenic signature and brown adipocyte apoptosis [61,70,76,77]. On this basis, and since mitochondria are widely recognized as a therapeutic target for melatonin, it is reasonable to speculate that melatonin might protect against obesity-induced BAT dysfunction by improving the performance of mitochondrial respiratory and antioxidant functions. In support of this speculation, our previous study has shown that chronic oral administration of melatonin improved the functionality of mitochondria isolated from iBAT of the ZDF rats by increasing mitochondrial respiration and reducing mitochondrial oxidative/nitrosative stress and susceptibility to apoptosis [29]. Mitochondria are dynamic organelles that continuously divide and fuse, and the mitochondrial fission and fusion process plays a key role in providing a basis for mitochondrial functions, including respiration and oxidative stress balance [7]. Mitochondrial fusion is believed to be beneficial because it increases mitochondrial efficiency, whereas mitochondrial fission seems to be associated with mitochondrial malfunctions and oxidative stress [7]. Given the potentially critical role of mitochondrial dynamics in regulating BAT thermogenesis [38,39,40,78,79], we elected to investigate the possible dose-dependent effect of melatonin on mitochondrial dynamics by Western blotting of OPA1 fusion and DRP1 fission markers. We found that both OPA1 and DRP1 were modulated by melatonin treatment in a dose-dependent fashion. Of note, the protein level of OPA1 increased significantly at all three doses of melatonin and reached the highest level at 10 mg/kg, whereas that of DRP1 significantly decreased at all three doses as well, with the lowest level equally being observed at doses of 1 and 10 mg/kg. This suggests that the balance between fusion and fission processes might be shifted toward mitochondrial fusion to maintain high-quality functional mitochondria, which can be essential to support efficient fatty acid β-oxidation and uncoupled mitochondrial respiration required for the maintenance of the brown adipocyte phenotype. Consistent with this suggestion, an earlier study in OPA1 BAT knockout mice has demonstrated that OPA1 deficiency impaired BAT activation and led to mitochondrial bioenergetic deficiency and thermogenic gene program downregulation, providing direct evidence that OPA1 plays a pivotal role in BAT thermogenic activation [40]. In vivo evidence in mice lacking the ATP-independent metalloprotease OMA1, which plays an essential role in the proteolytic inactivation of OPA1, reinforces the importance of OPA1-related fusion for BAT thermogenesis [39]. In addition, knockout models of BAT fusion-related markers in mice have been shown to lead to dysfunctional mitochondria and impaired BAT thermogenic activity [41,79]. Based on these findings, our results might implicate the fusion process as an important target for melatonin to enhance mitochondrial functions and preserve the BAT thermogenic signature and hence favour the maintenance of the BAT thermogenic phenotype under the deleterious conditions of obesity. The ability of melatonin to regulate mitochondrial dynamics is in line with other reports in different tissues under various pathological conditions [31,80,81,82,83]. On the other hand, these results imply that the increased fusion and decreased fission through melatonin treatment results in the inhibition of mitochondrial fragmentation. Excessive mitochondrial fragmentation due to imbalanced mitochondrial dynamics has often been reciprocally linked to mitochondrial oxidative stress, which is critical for inducing cell death [5,7]. We have previously analysed the mitochondrial fractions of ZDF rats’ iBAT for superoxide dismutase (SOD) activity and nitrite level as markers of oxidative stress and mitochondrial permeability transition pore (mPTP) activity as a marker of apoptosis [29]. The results have shown reduced SOD activity associated with increased nitrite content and mPTP activity in untreated ZDF rats compared with their lean littermates [29]. Interestingly, treatment with melatonin has been found to reduce the mitochondrial nitrosative and oxidative status and susceptibility to apoptosis by decreasing nitrite levels, increasing SOD antioxidant enzyme activity and inhibiting mPTP activity [29]. Given the close interaction between redox status and mitochondrial dynamics [7], we were prompted to ask whether melatonin protects against iBAT thermogenic dysfunction via direct interaction with components of mitochondria fusion-fission pathways, as the beneficial role of melatonin in promoting fusion and inhibiting fission processes may not be the cause, but rather the consequence of its protective effects in reducing mitochondrial oxidative stress. Consistent with this idea, SOD2 has been reported to enhance the mitochondrial fusion process independent of its antioxidant activity [84]. Therefore, given that melatonin is acknowledged to have direct antioxidative protection, including directly scavenging free radicals and enhancing the activity of the antioxidant enzyme [23], it is likely that melatonin promoted OPA1-mediated mitochondrial fusion by activating the antioxidant pathway. Though the precise molecular mechanisms through which melatonin regulates OPA1 and DRP1 protein levels were beyond the scope of this study, our findings provide preliminary insight into the role of melatonin in the regulation of BAT mitochondrial fission-fusion, which may play a key role in the deregulation of BAT thermogenic function in obesity and consequently could be novel future therapeutic targets.

The potential effect of melatonin on mitochondrial dynamics could not rule out the possibility that melatonin might restore the BAT thermogenic ability by recruiting thermogenic brown adipocytes through differentiation of brown adipocyte progenitors, given that progenitor cells have been found to still exist in obese BAT [85]. In support of this possibility, sheep brown adipocyte precursor cells isolated from perirenal BAT exposed to in vitro melatonin treatment have promoted brown adipocyte formation and increased protein expression levels of brown adipogenic markers via AMK-activated protein kinase, thus showing melatonin involvement in brown adipocyte differentiation [86]. Melatonin administration may promote the recruitment of new brown adipocytes, possibly by acting centrally on MT1 receptors located on neurons of the suprachiasmatic nucleus to increase the sympathetic tone in BAT, or directly on melatonin receptors located on the brown adipocytes (MT1 and MT2), which results in a reduction of intracellular cAMP levels with a lowering of PKA activity and cAMP-response element-binding protein phosphorylation [4,24].

The present findings, if confirmed clinically, suggest melatonin would be advantageous over other drugs that are currently being tested in clinical trials or EMA- and FDA-approved drugs that have now been extended to treat obesity, by targeting BAT but without serious adverse events on the one hand due to its safety profile, and, on the other hand, due to the fact that melatonin may act by directly targeting mitochondria rather than by secondary off-target effects [17]. 

## 5. Conclusions

The results presented here reveal for the first time that chronic oral administration of melatonin to obese diabetic ZDF rats at doses of 0.1, 1, or 10 mg/kg b.w over 6 wk enhances the BW gain and iBAT thermogenic features in a dose-dependent manner, with the underlying mechanism for the iBAT thermogenesis possibly being through restoring brown adipocyte mitochondrial function, which could have great therapeutic value, especially for promoting BAT reactivation among obese individuals with blunted BAT activity. Therefore, to translate these promising findings into potential clinical anti-obesity therapy, we estimate that the therapeutically effective dose for promoting BAT thermogenesis and, consequently, weight loss is within the dose range of 12 to 120 mg orally per day for a 75 kg adult individual, based on human equivalent dose calculation. Future powered randomized clinical trials are warranted to confirm this. In addition, future in vitro and in vivo in-depth studies are required to fully understand the potential effect of melatonin on iBAT mitochondrial dynamics and decipher the related underlying molecular mechanism, as a full understanding of the mechanisms could facilitate the clinical application of melatonin for obesity.

## Figures and Tables

**Figure 1 antioxidants-11-01646-f001:**
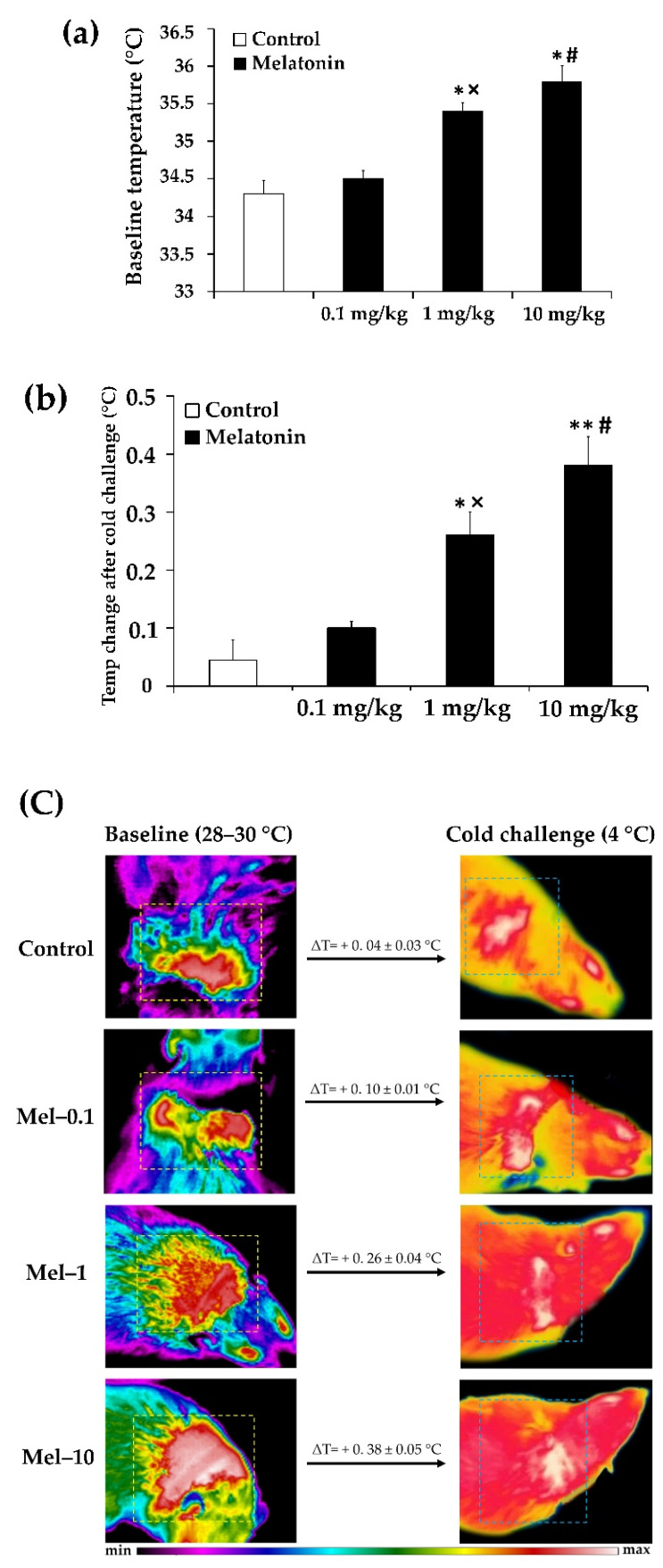
Dose-dependent effects of melatonin on iBAT thermogenesis in Zücker diabetic fatty rats as measured by infrared thermography. (**a**,**b**) Quantification of the interscapular skin temperature at baseline (**a**) and after acute cold challenge (**b**). (**c**) Representative infrared thermal images of the interscapular skin temperature at baseline and after acute cold challenge. Mel-0.1, Mel-1, and Mel-10: melatonin at a dose of 0.1, 1, and 10 mg/kg b.w, respectively. Values are means ± S.D (*n* = 4). * *p* < 0.05, ** *p* < 0.01 Mel-1 and Mel-10 vs. Control; # *p* < 0.05 Mel-10 vs. Mel-0.1; × *p* < 0.05 Mel-1 vs. Mel-0.1 (One-way ANOVA with Tukey post hoc test).

**Figure 2 antioxidants-11-01646-f002:**
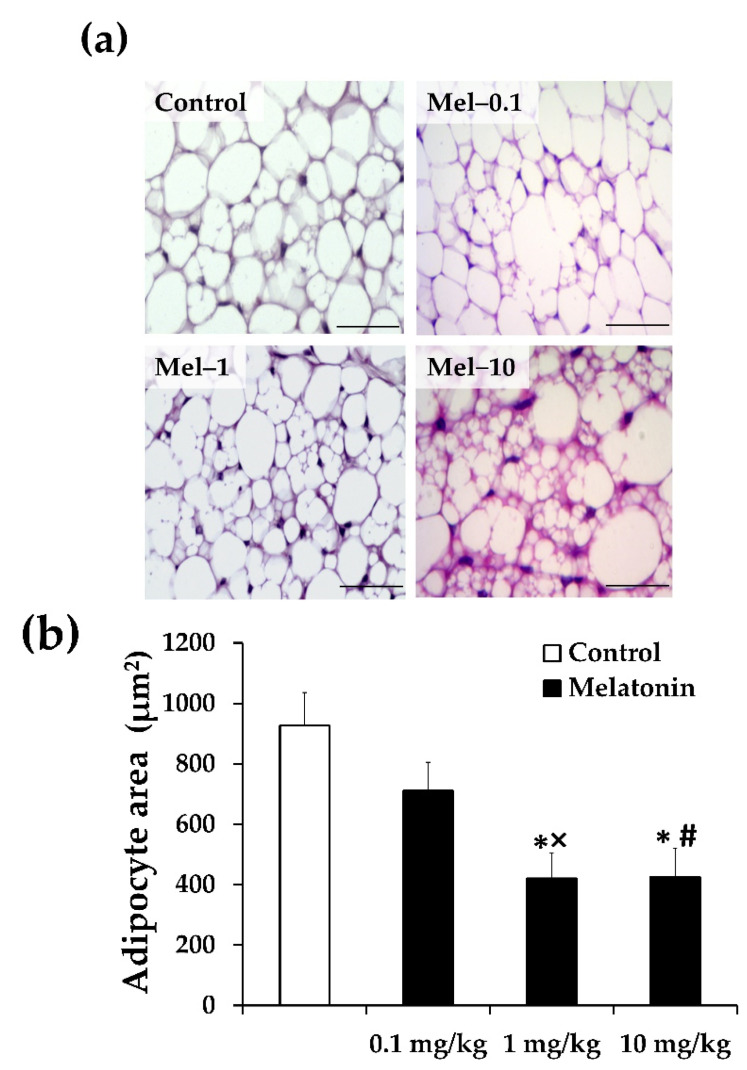
Dose-dependent effects of melatonin on iBAT morphology in Zücker diabetic fatty rats. (**a**) Representative haematoxylin and eosin staining of iBAT sections (original magnification ×400). (**b**,**c**) Mean adipocyte area (**b**) and frequency distribution of adipocyte area (**c**) were measured using a quantitative morphometric method with Image J software. Mel-0.1, Mel-1, and Mel-10: melatonin at a dose of 0.1, 1, and 10 mg/kg b.w, respectively. Values are means ± S.D (*n* = 200 adipocytes/group). * *p* < 0.05, ** *p* < 0.01, *** *p* < 0.001 Mel-0.1, Mel-1, and Mel-10 vs. Control; # *p* < 0.05 Mel-10 vs. Mel-0.1; × *p* < 0.05 Mel-1 vs. Mel-0.1 (One-way ANOVA Tukey with post hoc test). Scale bar: 50 μm.

**Figure 3 antioxidants-11-01646-f003:**
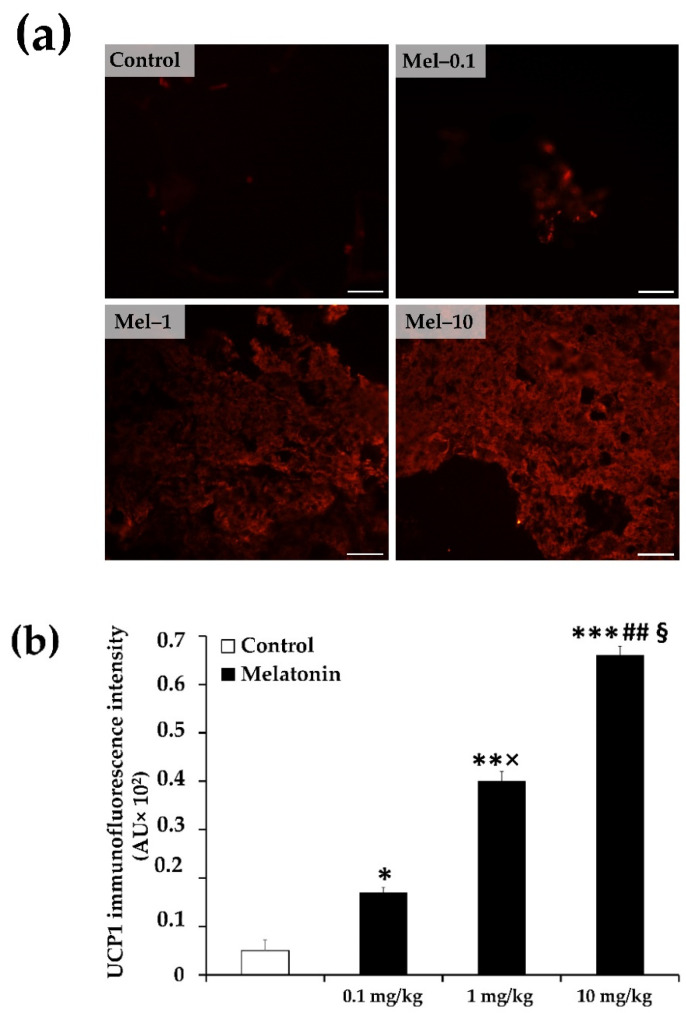
Dose-dependent effects of melatonin on hallmark thermogenic and mitochondrial dynamic markers in iBAT of Zücker diabetic fatty rats. (**a**) Representative microscopic immunofluorescent images of uncoupled protein 1 (UCP1) visualized with cyanine3 secondary antibody (original magnification ×200). (**b**) Immunofluorescence intensity quantification of UCP1. (**c**–**e**) Western blot densitometry quantification of UCP1, optic atrophy protein type 1 (OPA1), and dynamin-related protein1 (DRP1). (**f**) Representative Western blot images of UCP1, OPA1, and DRP1. Mel-0.1, Mel-1, and Mel-10: melatonin at a dose of 0.1, 1, and 10 mg/kg b.w, respectively. Values are means ± S.D (*n* = 3) of UCP1 fluorescence intensity and ratios of specific protein levels to β-actin (loading protein). * *p* < 0.05, ** *p* < 0.01, *** *p* < 0.001 Mel-0.1, Mel-1 and Mel-10 vs. Control; # *p* < 0.05, ## *p* < 0.05 Mel-10 vs. Mel-0.1; ^§^ *p* < 0.05 Mel-10 vs. Mel-1; × *p* < 0.05 Mel-1 vs. Mel-0.1 (One-way ANOVA with Tukey post hoc test). Scale bar: 50 μm.

**Table 1 antioxidants-11-01646-t001:** Dose-dependent effects of melatonin on body weight in Zücker diabetic fatty rats.

Groups	Control	Mel-0.1	Mel-1	Mel-10
Final body weight (g)	495.7 ± 6.4	473.4 ± 4.6	443.0 ± 3.1 *^×^	436.1 ± 8.5 *^#^
Body weight gain (%)	74.7 ± 1.7	66.3 ± 2.4	55.2 ± 1.1 *^×^	54.7 ± 1.8 *^#^

Mel-0.1, Mel-1, and Mel-10: melatonin at a dose of 0.1, 1, and 10 mg/kg b.w, respectively. Values are means ± S.D (*n* = 4). * *p* < 0.05 Mel-1 and Mel-10 vs. Control; # *p* < 0.05 Mel-10 vs. Mel-0.1; × *p* < 0.05 Mel-1 vs. Mel-0.1 (One-way ANOVA with Tukey post hoc test).

## Data Availability

Not applicable.

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
