# Peer review of "Dose-Dependent Effect of Melatonin on BAT Thermogenesis in Zücker Diabetic Fatty Rat: Future Clinical Implications for Obesity"

_antioxidants, 2022, doi:10.3390/antiox11091646_

Round 1

Reviewer 1 Report

The authors investigated the effect of melatonin on thermogenesis and body weight of BAT in obese diabetic rats. The following additional data is required:

1) qPCR results for mRNA expression levels of UCP1, OPA1, and DRP1

2) IHC results for protein expression of UCP1, OPA1, and DRP1 in iBAT tissues

3) The authors should elucidate whether melatonin directly regulates mitochondrial fusion and fission dynamics in iBAT.

4) The authors should investigate whether melatonin is directly involved in the regulation of the expression of OPA1 and DRP1 in iBAT, and whether these two molecules are key factors in the thermogenic effect of melatonin.

Reviewer 2 Report

This manuscript is on melatonin effects in diabetic rats. Authors clarified that melatonin treatment enhances sevral mitochondrial functions of interscapular brown fat cells in obese-diabetic rats in the previous report (Antioxidants 2021, 10, 1482). In this manuscript, you showed effects of melatonin on brown adipose tissue (BAT) was in dose dependent manner in obese rats. The effect of melatonin administration on BAT functions was analyzed in detail. BAT thermogenic repression in obesity owing to impairment of mitochondrial respiratory and antioxidant functionis discussed well (line 5-4-506). As a result, mitochondria are widely recognized as a therpapeutic target for melatonin, and effect of melatonin administration at low dose is clarified in this manuscript.   

Reviewer 3 Report

The article deals with an epidemiologically important topic - the increasing percentage of obese people. Considering the numerous health problems associated with obesity, it is very valuable to find new ways to fight this problem. Considering the pharmacological safety of melatonin, the use of this compound for weight loss is interesting.

The introduction gives a clear overview of the reasons the Authors were guided by undertaking the research subject and clearly presents the research hypothesis.

The research methods used by the Authors are properly selected and described in detail. The course of the described experiments does not raise any methodological and ethical doubts.

The statistical methods seem to be chosen appropriately.

The results verify the hypothesis at several levels of the organization of living matter (tissue, cell, molecular) and and suggest the mechanism of the observed changes.

The discussion is written very well, especially interesting are the fragments discussing the clinical possibilities of using the results presented in the paper.

From minor remarks I can point out some typographic flaws, like µm2 instead of µm2 in the description of the Y axis in Fig. 2b. They will certainly be corrected during the revision. I may also suggest not to use markers on the X axis when the data is presented in bar graphs. The markers then don't make sense.

Round 2

Reviewer 1 Report

The authors investigated the effect of melatonin on thermogenesis and body weight of BAT in obese diabetic rats. However, mechanistic studies of the role of melatonin in iBAT are lacking in this study.

Author Response

Please see attached our response.
